# A cloud algorithm based on the $O_2$-$O_2$ 477 nm absorption band featuring an advanced spectral fitting method and the use of surface geometry-dependent Lambertian-equivalent reflectivity

Alexander Vasilkov[1], Eun-Su Yang[1], Sergey Marchenko[1], Wenhan Qin[1], Lok Lamsal[2], Joanna Joiner[3], Nickolay Krotkov[3], David Haffner[1], Pawan Bhartia[3], and Robert Spurr[4]

[1]Science Systems and Applications Inc., Lanham, MD, USA
[2]Universities Space Research Association, Columbia, MD, USA
[3]NASA Goddard Space Flight Center, Greenbelt, MD, USA
[4]RT Solutions Inc., Cambridge, MA, USA

**Correspondence:** A. Vasilkov
(alexander.vasilkov@ssaihq.com)

**Abstract.** We discuss a new cloud algorithm that retrieves an effective cloud pressure, also known as cloud optical centroid pressure (OCP), from oxygen dimer ($O_2$-$O_2$) absorption at 477 nm after determining an effective cloud fraction (ECF) at 466 nm, a wavelength not significantly affected by trace gas absorption and rotational-Raman scattering. The retrieved cloud products are intended for use as inputs to the operational nitrogen dioxide ($NO_2$) retrieval algorithm for the Ozone Monitoring Instrument (OMI) flying on the Aura satellite. The cloud algorithm uses temperature-dependent $O_2$-$O_2$ cross-sections and incorporates flexible spectral fitting techniques that account for specifics of the surface reflectivity. The fitting procedure derives $O_2$-$O_2$ slant column densities (SCD) from radiances after $O_3$, $NO_2$, and $H_2O$ absorption features have been removed based on estimates of the amounts of these species from independent OMI algorithms. The cloud algorithm is based on the frequently used Mixed Lambert-Equivalent Reflectivity (MLER) concept. A geometry-dependent Lambertian-equivalent reflectivity (GLER) is used for the ground reflectivity in our implementation of the MLER approach. The OCP is derived from a match of the measured $O_2$-$O_2$ SCD to that calculated with the MLER method. Temperature profiles needed for computation of vertical column densities are taken from the Global Modeling Initiative (GMI) model. We investigate the effect of using GLER instead of climatological LER on the retrieved ECF and OCP. For evaluation purposes, the retrieved ECFs and OCPs are compared with those from the operational OMI cloud product which is also based on the same $O_2$-$O_2$ absorption band. Impacts of the application of the newly developed cloud algorithm to the OMI $NO_2$ retrieval are discussed.

## 1 Introduction

Satellite ultraviolet and visible (UV/Vis) nadir backscattered sunlight trace-gas algorithms need ac- curate estimates of cloud parameters in order to produce high quality data sets. Because of complexity of cloud effects on the radiation field in the atmosphere, clouds in trace-gas algorithms are treated using multiple simplifying assumptions. Among them, the fundamental assumptions are (1) the independent pixel approximation (IPA) that neglects the horizontal transport of radiative energy

between the clear-sky and overcast subpixels; and (2) the assumption of horizontally and vertically homogeneous clouds that substantially simplifies radiative transfer in the clouds. For trace-gas retrievals it is important to estimate the photon pathlengths in the atmosphere that determine trace-gas absorption and thus affect the measured top-of-atmosphere (TOA) radiance. The photon pathlengths in a cloudy atmosphere are determined by the following most important cloud parameters: the geometrical cloud fraction and the cloud vertical extinction profile (Stammes et al., 2008). Other cloud parameters: the cloud phase, the cloud particle shape, and the particle size distribution that determine the cloud phase scattering function are usually not considered. Because of the limited informational content of TOA radiances, all of those parameters cannot be independently retrieved from the radiance measurements. That is why the additional cloud assumptions should be adopted. For instance, if we retrieve the cloud parameters from an oxygen absorption band and assume a model of scattering cloud (Loyola et al., 2017), we have to adopt a priori values of the cloud microphysical parameters and cloud vertical extent assuming a homogeneous cloud layer and to add information on cloud fraction from other measurements.

One of the simplest cloud models is the so-called mixed Lambertian-equivalent reflectivity (MLER) model. Many trace-gas algorithms are based on the MLER model. For example, the MLER model is currently used in most trace-gas (Veefkind et al., 2006; Bucsela et al., 2013) and cloud (Acarreta et al., 2004; Joiner and Vasilkov, 2006; Veefkind et al., 2016) retrieval algorithms for the Ozone Monitoring Instrument (OMI) (Levelt et al., 2006), a Dutch/Finnish UV/Vis sensor flying on the National Aeronautics and Space Administration (NASA) Aura satellite. For each field of view (FOV) the MLER model treats cloud and ground as horizontally homogeneous, opaque Lambertian surfaces and mixes them using IPA. According to the IPA, the measured TOA radiance is a sum of the clear sky and overcast subpixel radiances that are weighted with an effective cloud fraction ($f$) (e.g., Stammes et al., 2008). $f$ is calculated by inverting the equation

$$I_m = I_g(R_g)(1-f) + I_c(R_c)f \qquad (1)$$

at a wavelength not substantially affected by rotational-Raman scattering (RRS) or atmospheric absorption, where $I_m$ is the measured TOA radiance, $I_g$ and $I_c$ are the precomputed clear sky (ground) and overcast (cloudy) subpixel TOA radiances, and $R_g$ and $R_c$ are the corresponding ground and cloud Lambertian-equivalent reflectivities (LER), respectively.

The MLER model typically assumes $R_c = 0.8$ (McPeters et al., 1996; Koelemeijer et al., 2001). This assumption more accurately accounts for Rayleigh scattering in partially cloudy scenes (Ahmad et al., 2004; Stammes et al., 2008) and also accounts for scattering/absorption that occurs below a thin cloud. In this paper we also adopt $R_c = 0.8$ for the $O_2$-$O_2$ cloud algorithm. The ground reflectivity is usually taken from an LER or surface albedo climatology derived from satellite observations (e.g., Herman and Celarier, 1997; Kleipool et al., 2008). In reality, reflection of incoming direct and diffuse solar light from land or ocean surfaces is sensitive to the sun–sensor geometry. This dependence is described by the bidirectional reflectance distribution function (BRDF). Impact of surface BRDF on the cloud and trace-gas retrievals has been studied since 2010. For instance, Zhou et al. (2010) reported that accounting for surface BRDF effects can change $NO_2$ retrievals by up to 20%. The newest advances in this field and more references can be found in the latest paper by Lorente et al. (2018). To account for the BRDF, we developed a new model of geometry-dependent Lambertian equivalent reflectivity (GLER) that was implemented within

the existing OMI cloud and $NO_2$ retrieval algorithms (Vasilkov et al., 2017). This implementation required changes only to the input surface reflectivity database, thus simplifying the introduction of BRDF effects.

The MLER model compensates for photon transport within a cloud by placing the Lambertian surface somewhere in the middle of the cloud instead of the top (Vasilkov et al., 2008). As clouds are vertically inhomogeneous, the pressure of this surface does not necessarily correspond to the geometrical center of the cloud, but rather to the so-called optical centroid pressure (OCP) (Vasilkov et al., 2008; Sneep et al., 2008; Joiner et al., 2012). The cloud OCP can be thought of and modeled as a reflectance-averaged pressure level reached by back-scattered photons (Joiner et al., 2012). Cloud OCPs are the appropriate quantity for use in trace-gas retrievals from satellite instruments (Vasilkov et al., 2004; Joiner et al., 2006, 2009). Cloud top pressures derived from thermal IR measurements are not equivalent to OCPs and do not provide good estimates of solar photon pathlengths through clouds that are needed for trace-gas retrievals from UV/Vis backscatter measurements (Joiner and Vasilkov, 2006; Vasilkov et al., 2008; Joiner et al., 2012).

The main goal of this paper is to document new approaches to the development of a cloud algorithm based on the $O_2$-$O_2$ absorption band at 477 nm (Yang et al., 2015). These approaches include an advanced spectral fitting algorithm for TOA radiances and the use of surface GLERs to replace climatological LER data sets. This new cloud algorithm is intended for the operational OMI $NO_2$ algorithm and is planned as a backup cloud algorithm for the upcoming Tropospheric Emissions: Monitoring Pollution (TEMPO) geostationary mission (Zoogman et al., 2017). Our spectral fitting procedure is similar to that developed by Marchenko et al. (2015) for $NO_2$ SCD retrieval. It relies on the temperature-dependent $O_2$-$O_2$ cross-sections (Thalman and Volkamer, 2013) and derives the $O_2$-$O_2$ SCD using $O_3$, $NO_2$, and $H_2O$ slant column estimates from independent OMI algorithms. This is an implementation choice that is designed to minimize potential errors due to interference between $O_3$, $NO_2$, and $O_2$-$O_2$ spectral absorption features during the fitting procedure. We apply the new cloud algorithm to the OMI $NO_2$ retrievals and show $NO_2$ column changes related to the use of the new cloud algorithm.

## 2 Data and Methods

### 2.1 OMI and MODIS data

We use several data sets from the OMI and MODerate resolution Imaging Spectroradiometer (MODIS) instruments flying on the NASA Aqua and Terra satellites. OMI is a spectrometer that acquires Earth and solar spectra at UV/Vis wavelengths from 270–500 nm with a spectral resolution of approximately 0.5 nm. The OMI ground footprint varies; near nadir, it is approximately 12 km along the satellite track and 24 km across the 2600 km track. The footprint size increases towards the swath edge. We use TOA radiance and solar irradiance in the OMI Vis channel to retrieve cloud parameters and $NO_2$ amounts.

The MODIS-derived BRDF kernel coefficients from the 16-day MCD43GF data set (Schaaf et al., 2011) are used to compute GLERs over land for the OMI swath (Vasilkov et al., 2017). The kernel coefficients are provided for snow-free land and permanent ice at a high spatial resolution. Over transient snow-covered regions, we retain the standard climatological LER of Kleipool et al. (2008) that was routinely used for the previous cloud retrievals.

### 2.1.1 GLER computation

The BRDF kernel coefficients are averaged over an OMI field-of-view and used to calculate the TOA radiance for a given observational geometry assuming pure Rayleigh scattering in the atmosphere. For radiative transfer (RT) calculations, we use the Vector Linearized Discrete Ordinate Radiative Transfer (VLIDORT) code (Spurr, 2006). VLIDORT computes the Stokes vector in a plane-parallel atmosphere with a Lambertian or non-Lambertian underlying surface. It has the ability to deal with attenuation of solar and line-of-sight paths in a spherical atmosphere, which is important for large solar zenith angles (SZA) and viewing zenith angles (VZA). VLIDORT accounts for polarization at the ocean surface using a full Fresnel reflection matrix.

The TOA radiance computed by VLIDORT is then inverted to derive GLER using the following exact equation:

$$I_{TOA} = I_0 + \frac{RT}{1 - RS_b}, \tag{2}$$

where $I_0$ is the TOA radiance calculated for a black surface, $R$ is the GLER, $T$ is the total (direct + diffuse) solar irradiance reaching the surface converted to the ideal Lambertian-reflected radiance (by dividing by $\pi$) and then multiplied by the transmittance of the reflected radiation between the surface and TOA in the direction of a satellite instrument, and $S_b$ is the diffuse flux reflectivity of the atmosphere for the case of its isotropic illumination from below (Dave, 1978). All quantities, $I_0$, $T$, and $S_b$ are calculated using a known surface pressure. We use a monthly climatology of surface pressure taken from the Global Modeling Initiative (GMI) chemistry transport model driven by the NASA GMAO GEOS-5 global data assimilation system (Rienecker et al., 2011) with spatial resolution of $1°$ latitude by $1.25°$ longitude. Surface pressure for each OMI pixel $P_s$ is calculated as follows:

$$P_s = P_s(\text{GMI})exp(-\Delta z/H), \tag{3}$$

where $P_s(\text{GMI})$ is the GMI surface pressure at resolution of $1° \times 1.25°$, $\Delta z = z - z(\text{GMI})$, $z$ is the terrain height of the OMI pixel from a digital elevation model (DEM), $z(\text{GMI})$ is the terrain height at resolution of $1° \times 1.25°$, $H = (kT)/(Mg)$ is the scale height, where $k$ is Boltzmann constant, $T$ is the GMI air temperature at the surface, $M$ is the mean molecular weight of air, and $g$ is the acceleration due to gravity.

To calculate TOA radiance over water surfaces, we account both for light specularly reflected from a rough water surface and also for diffuse light backscattered by water bulk and transmitted through the water surface. Reflection from the water surface is described by the Cox-Munk slope distribution function as implemented in Mishchenko and Travis (1997). Diffuse light from the ocean is calculated using a Case 1 water model that has chlorophyll concentration as a single input parameter. Bidirectionality of the underwater diffuse light is accounted for following Morel and Gentili (1996).

More details about the GLER computation can be found in Vasilkov et al. (2017). An important update of our ocean surface model is the use of a variable wind speed instead of a single climatological wind speed of $5\,\text{m/s}$ as in Vasilkov et al. (2017). Retrievals of wind speed are taken from the Advanced Microwave Scanning Radiometer for the Earth Observing System (AMSR-E) that flies on NASA's Aqua satellite (with Aura/OMI closely following Aqua). The use of wind speed from the

AMSR-E measurements improves in the GLER over ocean. Thanks to the higher spatial resolution of AMSR-E, it is possible to match fine structure of the wind field to the TOA radiances and GLERs over the sun glint affected areas.

## 2.2 The $O_2$-$O_2$ slant column density fitting algorithm

The operational OMI $O_2$-$O_2$ SCD retrieval (Acarreta et al., 2004; Veefkind et al., 2016) uses the Differential Optical Absorption Spectroscopy (DOAS) (Platt and Stutz, 2006) approach, simultaneously retrieving SCDs of $O_2$-$O_2$ and $O_3$ in the 460-490 nm wavelength interval and using single-temperature $O_2$-$O_2$ cross-sections and a first-degree polynomial approximating the wavelength dependence of reflectances in the fitting window.

Here, we generally follow the approach developed by Marchenko et al. (2015) for the $NO_2$ SCD estimates. Instead of simultaneous retrieval of coefficients of multiple parameters as takes place in the classical DOAS formalism, we divide the problem into a series of sequential steps (Fig. 1).

Step 1 involves the removal of interfering trace-gas absorption. The spectral range chosen for the $O_2$-$O_2$ SCD retrievals is affected by relatively strong $O_3$ absorption that, in most cases, distorts the $O_2$-$O_2$ profiles (Fig. 2). The same applies to $NO_2$ absorption over polluted regions (e.g., the Beijing area, see Fig. 3) and, to a far lesser extent, the mainly equatorial regions over open-water Pacific, where the $H_2O$ absorption may distort the flanks of the broad $O_2$-$O_2$ profiles (Fig. 2). Note the clear presence of the ozone feature around $\lambda \sim 462$ nm, as well as the large distortion of the $O_2$-$O_2$ profile caused by the broad ozone absorption around $\lambda \sim 482$ nm. In this particular example, the only easily recognizable Ring-spectrum feature is seen at $\lambda \sim 486.5$ nm. The gradual $\sim 13\%$ change in reflectances between 450 and 500 nm comes from a combination of comparable-strength signals: the ozone absorption and the Rayleigh-scattering component. For better guidance, in Figs. 2– 5 we show scaled absorption spectra of the main trace gases that may contribute to the general appearance of a reflectance spectrum. We keep the same plotting style for Figs. 2– 5, though noting that in each particular example one may see quie different impact from the same absorption constituent. E.g., while the $NO_2$-related signal barely registers in Fig. 2 (practically unpolluted region), the heavily contaminated Beijing area (Fig. 3) shows clear presence of the $NO_2$ absorption at $\lambda \sim 457$–466 nm (3 features), $\lambda \sim 475$ and 480 nm (these two are superposed on the broad $O_2$-$O_2$ absorption), as well as the well-defined $NO_2$ absorption at $\lambda \sim 489$ nm that rivals the strength of the retrieved $O_2$-$O_2$ feature. In this particular case of the heavily $NO_2$-polluted region the omnipresent $O_3$ absorption plays far less important role compared to the spectrum shown in Fig. 2. In addition, though to a far lesser extent (when compared to ozone, Ring and $NO_2$), the $H_2O$ absorption may distort the flanks of the broad $O_2$-$O_2$ profiles, mainly in the equatorial regions over open-water Pacific.

The spectral domain chosen for the $O_2$-$O_2$ retrieval is not optimal for simultaneous $O_3$, $NO_2$, or $H_2O$ estimates. Optimal fitting windows are the $\sim 290$–340, 400–465 and 435–450 nm intervals for $O_3$, $NO_2$, and $H_2O$, respectively. Hence, to minimize the biases that may be introduced by the sub-optimal DOAS estimates of the interfering trace-gas species, we use the SCDs provided by independent OMI products: $NO_2$ and $H_2O$ from OMNO2SCD (Marchenko et al., 2015) and $O_3$ from OMDOAO3 (Veefkind et al., 2006), and remove the corresponding absorption features from the observed radiances. We find that, as expected, at large solar-zenth angles the corrections based on the UV $O_3$ SCDs retrievals result in large spectral residuals pointing to systematic underestimates of $O_3$ absorption strength. This stems from the notion that the relatively (to the

visual range) higher Rayleigh optical depth effectively masks the lower-atmosphere $O_3$ absorption. Hence, at SZA $> 80°$ we adjust the UV SCDs by a constant 1.25 coefficient. This helps to reduce the spectral residuals related to the underestimated $O_3$ absorption to a manageable (on average, $< 0.1\%$) level.

Step 2 closely follows the approach from Marchenko et al. (2015), comprising the simultaneous, iterative wavelength adjust-
ment and Ring spectrum removal. At each FOV (row, 60 in total) the reflectances are produced from the individual, pre-filtered earthshine radiances (from Step 1) normalized by the monthly-averaged OMI irradiances. These irradiances are iteratively adjusted to accommodate slight relative (radiances vs. irradiances) wavelength shifts. For estimates of the line-filling factors(i.e. the Raman scattering amplitudes), we use appropriate combination of the air and water Raman scattering spectra (Vasilkov et al., 2002). We split the retrieval region in two "micro-windows", 451-469 and 483-496 nm, and iteratively evaluate the wave-
length shifts and the Raman-spectrum amplitudes in each window. For the final removal of the Raman scattering patterns, we use an average of the two "micro-window" estimates. The individual "micro-window" wavelength shifts are used for wave-
length adjustments of irradiances in each "micro-window", interpolating these estimates in the 469-483 nm domain occupied by the main $O_2$-$O_2$ absorption.

Step 3 involves normalizing the $O_2$-$O_2$ profile in preparation for SCD evaluations. We deem this step to be the most important
procedure; it may change the outcome by as much as $\sim20\%$ in extreme cases such as the open-water scenes(Fig. 4) and Sahara desert (Fig. 5) representing two extremes and the remaining cases falling in-between. In both figures, the upper panels show the observed reflectances before (black lines) and after (blue lines) the removal of trace-gas absorption and the Raman line-filling patterns, and the red lines follow the adopted continuum fits. Note the profound difference between the wavelength dependencies of the reflectancies in these extreme cases. While the cloud-free, open-water case (Fig. 4) is predominantly
Rayleigh-controlled, leading to a steep decline in reflectances, the much brighter Sahara surface controls the appearance of the radiances at long OMI wavelengths, leading to the gradual increase in reflectances (Fig. 5). The lower panels show normalized $O_2$-$O_2$ profiles. In a case-by-case study of the presumably cloud-free areas, we have found that various combinations of linear functions fitting the flanks of the $O_2$-$O_2$ profile lead to gross under-estimates (mainly over open-water areas) or over-estimates (deserts and semi-deserts) of the retrieved scene pressures that are directly linked to the biases in the SCD evaluations. Hence,
we have implemented a more flexible approach, defining two broad categories of the surface reflectances and applying different fitting approaches to each of them.

The reflectances from Step 2 are averaged in 2-nm intervals, providing a set of estimates at $\lambda =463$ and 495 nm that are partitioned into two general categories. The first broad category comprises all the relatively cloud-free low-reflectance scenes, with $\frac{r(463)}{r(495)} > 1.05$ and $r(463) < 0.25$. The second class includes the remaining scenes. For both categories, the fitting starts
from applying the 3rd-degree polynomial to the 459–466 and 484–494 nm regions, identifying and eliminating large ($\sim 5\sigma$, i.e., $\pm0.5\%$) deviations and then repeating the procedure, ultimately normalizing the reflectances in the 450–500 nm range by the fit. The normalized reflectances in the 459–465 and 484–490 nm intervals are re-fitted with a 1st-degree polynomial and, again, all reflectances in the 450–500 nm range are re-normalized by this fit. This concludes the fitting for the 2nd category of scenes. However, at this point the fitting proceeds for the 1st class of the relatively cloud-free, low-reflectance scenes. Yet
again, the normalized radiances in the 465–470 and 482–487 nm intervals are iteratively (rejecting the large $\pm1\%$ deviations)

re-fitted with a 2nd-degree polynomial, then this fit is applied exclusively to the region occupied by the $O_2$-$O_2$ profile, 465–491 nm. The line edges are further refined by applying piece-wise fits (1st or 2nd-degree polynomials) to the relatively narrow windows, 459–465, 486–491 nm, thus concluding this rather involved procedure for the first category of the low-reflectance scenes.

Step 4, the SCD retrieval, follows the approach described in Marchenko et al. (2015). Here, preliminary SCD values are obtained from two algorithms, the Nelder-Mead minimization method and the least-squares Levenberg-Marquardt fit (Press et al., 1992), taking the latter as a default and fitting the normalized (Step 3) $O_2$-$O_2$ profile in the 465–487 nm interval. These evaluations are repeated for each temperature-dependent $O_2$-$O_2$ cross-section; there are five of them measured by Thalman and Volkamer (2013). Each cross-section is fitted to the data, providing an individual root-mean-squared (RMS) value of the fitting residuals. These 5 RMS values are approximated by a parabolic function. The minimum of the function is used to construct via linear interpolation a synthetic $O_2$-$O_2$ profile that is removed from the normalized reflectances, thus leaving us with the residuals that are presumably dominated by instrumental noise. The noise is reduced by the iterative procedure similar to one described in Marchenko et al. (2015). The final SCD evaluation is performed over a slightly broadened wavelength range, 463–488 nm.

As implemented, the algorithm relies on optimal SCD retrievals of the $O_3$, $NO_2$ and $H_2O$ trace gases, as well as preliminary cloud-fraction estimates. The latter is used exclusively over deep-water areas during the wavelength calibration and the Raman scattering removal. If needed, such cloud fractions can be substituted for appropriately adjusted reflectances, thus vying for self-sufficiency. The use of independent $O_3$, $NO_2$ and $H_2O$ SCDs is an essential part of the algorithm that, especially for the scenes with heavy $O_3$ and $NO_2$ loads, leads to more accurate $O_2$-$O_2$ SCDs. The use of the trace-gas SCDs does not create any paradox when the $NO_2$ values would be used in order to retrieve cloud properties that should be incorporated into the $NO_2$ estimates. Note that in the implemented algorithm we use the $NO_2$ SCD estimates that can be obtained without any relevance on cloud properties. These cloud properties are used later, during the conversion of the $NO_2$ slant columns to the $NO_2$ vertical columns. Opting for a complete self-reliance of the cloud algorithm, one may substitute the required $O_3$, $NO_2$ and $H_2O$ SCDs for SCD estimates provided by the appropriate trace-gas climatologies.

## 2.3 Cloud algorithm

The $O_2$-$O_2$ cloud algorithm described here is based on the $O_2$-$O_2$ absorption band at 477 nm. This algorithm is broadly similar to the operational $O_2$-$O_2$ cloud algorithm developed at the Royal Meteorological Institute of the Netherlands (KNMI) known as OMCLDO2 (Acarreta et al., 2004; Sneep et al., 2008; Veefkind et al., 2016). However, our approach differs in a number of aspects.

First, we use normalized radiance at 466 nm to compute $f$ with equation (1) in a separate step. This wavelength was selected because it is not significantly affected by gaseous absorption and rotational-Raman scattering and it is still sufficiently close to the $O_2$-$O_2$ absorption band center at 477 nm. $f$ is calculated using linear interpolation of lookup tables (LUT) of $I_g$ and $I_c$. The tables were generated for 23 different surface/cloud pressures, 20 surface reflectivities, 30 SZAs, 20 VZAs, and 20 relative azimuth angles. Nodes and their locations were selected on a basis of the analysis of interpolation errors. A threshold for

acceptable interpolation error was set at 0.2%. It should be noted that aerosols are implicitly accounted for in the determination of $f$, as they are treated (like clouds) as particulate scatters.

Our algorithm retrieves cloud OCP from OMI-derived oxygen dimer SCD at 477 nm. The OCP, here also denoted as $P_c$, is estimated using the MLER method to compute the appropriate air mass factors (AMF) (Yang et al., 2015). To solve for OCP, we invert the following equation

$$\text{SCD} = \text{AMF}_g(P_s, R_g)\text{VCD}(P_s)(1 - f_r) + \text{AMF}_c(P_c, R_c)\text{VCD}(P_c)f_r, \tag{4}$$

where VCD is the vertical column density of $O_2$-$O_2$ (VCD=SCD/AMF), $\text{AMF}_g$ and $\text{AMF}_c$ are the precomputed (at 477 nm) clear sky (subscript $g$) and overcast (cloudy, subscript $c$) subpixel AMFs, $P_s$ is the surface pressure, and $f_r$ is the cloud radiance fraction (CRF) given by $f_r = fI_c/I_m$. Equation 4 is similar to that frequently used for retrieval of trace-gas VCDs with the MLER model provided $P_c$ and $R_c$ are known (see e.g. (Veefkind et al., 2006)). Here we use this equation for retrieval of $P_c$ assuming that the $O_2$-$O_2$ VCD is known. The CRF is calculated at 466 nm. CRF defines a fraction of TOA radiance reflected by the cloud. It should be noted that CRF is wavelength dependent (see discussion in Section 3.4.1). The CRF retrievals at different wavelengths are included in our output. Lookup tables of the TOA radiances and AMFs were generated using VLIDORT. Temperature profiles needed for estimation of VCD and AMF are taken from the NASA GMAO GEOS-5 global data assimilation system (Rienecker et al., 2011).

In addition to OCP, we retrieve the so-called scene pressure, $P_{sc}$. The scene pressure is derived from Eq. 4 assuming that $f_r = 1$ and $R_c$ is equal to the scene LER, $R_{sc}$:

$$\text{SCD} = \text{AMF}_c(P_{sc}, R_{sc})\text{VCD}(P_{sc}). \tag{5}$$

$R_{sc}$ is determined from the measured TOA radiance using Eq. 2 for a known surface pressure. In the absence of clouds and aerosols, the $P_{sc}$ should be equal to $P_s$. $P_{sc}$ is therefore an important diagnostic tool for evaluation of the performance of cloud pressure algorithms.

## 3 Results and Discussion

### 3.1 Evaluation of the cloud algorithm

To evaluate our cloud algorithm we have compared the retrieved values of $f$ and $P_c$ with those from the operational OMCLDO2 version 2 (Veefkind et al., 2016). For this comparison, the cloud products are retrieved for the climatological surface LER (Kleipool et al., 2008) identical to that used in OMCLDO2 v2. Figure 6 shows scatter plots of ECFs calculated with our algorithm versus those calculated with OMCLDO2 for land and ocean for a selected date, November 13, 2006. Different dates show very similar trends. The scatter of data around the 1:1 line is somewhat higher for low values of $f$s. The mean differences between the two data sets do not exceed 0.01 for all values of $f$. The standard deviation of the $f$ differences is within 0.01 for ocean and 0.03 for land. Differences in values of $f$ are probably due to contrasting approaches used in the two

algorithms. We retrieve the ECF at 466 nm independently from the OCP retrieval, whereas OMCLDO2 retrieves the ECF and OCP simultaneously at 477 nm.

Figure 7 shows scatter plots of OCPs calculated with our algorithm versus those calculated with OMCLDO2 v2 for the same day. There is a bias between OMCLDO2 and our algorithm: our OCP retrievals are higher than those from OMCLDO2 by about $50\,\text{hPa}$ on average. The standard deviation of the OCP differences ranges from about $100\,\text{hPa}$ for $\text{OCP} < 400\,\text{hPa}$ to $150\,\text{hPa}$ for lower OCPs. Higher OCP retrievals from our algorithm as compared to OMCLDO2 can be related to slightly higher $O_2$–$O_2$ SDC estimates and also to differences in ECF which affect the OCP retrievals.

Figure 8 shows a comparison of $P_s$ and $P_{sc}$ from our algorithm and OMCLDO2 v2 along cross track position 20 of the OMI orbit 4415. A similar comparison of the OMCLDO2 v2 $P_{sc}$ and with $P_s$ for this cross track position was carried out in Veefkind et al. (2016). We added our scene pressure and cloud fraction for this cross track position to this comparison. First, $P_{sc}$ retrievals from both algorithms agree very well with the surface pressures for the high reflectivity scenes in Greenland (OMI scan lines along the orbit with numbers iTimes=1300-1400). For mostly cloud free conditions over the ocean (iTimes $\sim 400$ and iTimes $\sim 270$), the scene pressures retrieved from both algorithms are higher than the surface pressures. These differences for those oceanic regions are slightly lower for our algorithm than for OMCLDO2. Over desert and semi-desert areas, $P_{sc}$ retrieved with our algorithm over mostly cloud free conditions (iTimes=720-900 and iTimes $\sim 570$) is close to the surface pressure while OMCLDO2 significantly overestimates the scene pressure. We attribute the better performance of our algorithm over deserts and semi-deserts to the special adjustment of the spectral fitting procedure for those areas (see Sec. 2.2 and Fig. 5). Figure 9 shows maps of differences between the scene pressure and the surface pressure retrieved from our algorithm and OMCLDO2 v2. The differences are shown for mostly clear scenes with $f < 0.25$. A comparison of the maps shows that the performance of our algorithm over land is better than the performance of OMCLDO2 v2. This is clearly seen for Sahara, the Arabic Peninsula, Australia, etc. Over the ocean, our algorithm performs slightly better only for certain areas in the Southern Ocean, e.g. for three locations at $\sim 55^\circ$S: $\sim 60^\circ$W, $\sim 20^\circ$W, and $\sim 100^\circ$E. Over other areas of the ocean, the performance of two algorithms is similar.

### 3.2 Comparison of ECFs derived with GLER and climatological LER

Figure 10 shows scatter plots of $f$ calculated with GLER versus those calculated with climatological LER (ClimLER) (Kleipool et al., 2008) in the form of 2-D histograms. The color scale on the 2-D histograms represents a number of data points. As expected, the scatter and systematic deviation of data around the 1:1 line diminishes with increasing $f$. Over land, $f$ calculated with GLER is mostly higher than $f$ calculated with the climatological LER (Fig. 10a) particularly for low $f$. This is explained by differences between GLERs and climatological LERs (Vasilkov et al., 2017) which have the most pronounced impact on $f$ for low cloudiness. The GLER values are mostly lower than the climatological LERs because the former are derived from atmospherically corrected MODIS radiances while the latter are affected by residual aerosols. Moreover, climatological LERs can be contaminated by clouds owing to substantially larger sizes of OMI pixels as compared with those of MODIS. As it follows from Eq. 1, lower surface LER leads to lower clear sky radiance, thus increasing $f$. For the most important for $NO_2$ retrieval range of $f < 0.25$, $f$ retrieved with GLER is higher than that retrieved with climatological LER by approximately

0.02 on average. The standard deviation of the GLER–ClimLER $f$ differences varies between 0.03-0.05 depending on the $f$ value.

Over ocean, the GLERs are higher than the climatological LERs in the areas affected by sun glint and areas observed at large viewing zenith angles (Vasilkov et al., 2017). Therefore, $f$ retrieved with GLER in these areas is lower than that retrieved with the climatological LER (Fig. 10b). For other oceanic regions, the GLERs are slightly lower than the climatological LERs resulting in slightly higher values of $f$. For the range of $f < 0.25$, which is frequently used in tropospheric trace gas retrievals, the mean difference between $f$ retrieved with GLER and that retrieved with climatological LER is approximately 0.02 when averaged globally. The standard deviation of the $f$ differences varies within 0.02-0.03 depending on the $f$ value. Even though the $f$ differences are small on average, they can be as large as 0.05-0.07 for individual pixels – this is quite significant for the low $f$ range. It should be noted that the fraction of negative $f$ retrievals is lower when using GLER as compared with the climatological LER. This is clear evidence of the improvement of $f$ retrievals with GLER.

Figure 11 is a geographic map of differences between $f$ calculated with GLER and climatological LER for November 13, 2006. Over the ocean, the most prominent are sun glint areas where the negative $f$ differences (GLER–ClimLER) are at maximum. This is because the climatological LERs are derived from minimum values of LERs from a long time series (up to 5 years) of observations over a given area; that is why the impact on observations affected by sun glint is somewhat mitigated. Over land, the $f$ differences are mostly positive due to aerosol and possible cloud contamination of the climatological LERs (note that the relatively large OMI footprints lead to predominance of cloudy scenes).

One indicator of the cloud algorithm performance is the cross-track dependence of the retrieved $f$ and OCP averaged over a given latitude bin. Ideally, this dependence should be fairly flat if the retrieved parameters are averaged globally. Figure 12 compares the cross-track dependencies of $f$ retrieved with GLER and ClimLER for Nov. 13, 2006. The comparison is carried out for $30°$ latitude bins for land and ocean separately and specifically for the low $f$ range $0.05 < f < 0.25$. Over land, the cross track dependence of $f$ is reasonably flat. $f$ retrieved with GLER is mostly lower than that retrieved with the climatological LERs because GLER is generally lower that the climatological LER. Over the ocean, the most noticeable feature of the cross track dependence is related to the sun glint area in the $30°S-0°$ latitude bin. Even though the use of GLER smooths out the irregularity in the cross-track dependence of $f$, it slightly overcorrects this irregularity owing to overestimation of GLER in sun glint areas. We have not yet determined the exact cause of this GLER overestimation. However, our preliminary radiative transfer simulations show that the presence of non-absorbing aerosol with low optical thickness ($\tau < 0.2$) can reduce the sun glint GLER. It should be noted that the cross track dependence of $f$ for cloudy scenes with $f > 0.25$ is much flatter than that for scenes with $f < 0.25$.

## 3.3 Comparison of OCPs produced with GLER and climatological LER

Figure 13 shows 2D histogram plots of OCPs calculated with GLER and climatological LER. Over land, the GLER-retrieved OCPs are slightly higher than those retrieved with climatological LER (Fig. 13a). Over ocean, the data are closer to the 1:1 line (Fig. 13b) but the overall scatter is somewhat higher than that over land. Data where the GLER-retrieved OCPs are lower than those retrieved with climatological LER are mostly from sun glint areas. There is a cluster of data where both GLER-retrieved

OCPs and those retrieved with climatological LER are higher than the surface pressure (1013 hPa). These data are retrieved over virtually clear sky conditions, and there are two possible reasons for this effect. First, OCP retrievals higher than the surface pressure are evidence of enhanced $O_2$-$O_2$ absorption. This can be caused by scattering, low-altitude aerosols in which enhanced photon pathlength prevails other aerosol effects. A second possible reason is due to an remaining deficiency in our spectral fitting procedure which somehow overestimates SCDs over the clear-sky ocean. A very small fraction of the OCP retrievals with values around 100 hPa is likely due to an artifact caused by the LUT extrapolation over the minimum pressure node of 100 hPa.

Geographic distribution of the differences between OCPs calculated with GLER and climatological LER is shown in Figure 14. Over ocean, most areas with negative differences (GLER - ClimLER) correspond to sun glint.

Figure 15 shows cross-track dependencies of OCPs retrieved with GLER and climatological LER for Nov. 13, 2006. Again, the range $0.05 < f < 0.25$ is used for the analysis. Over the ocean areas affected by sun glint, OCPs retrieved with GLER are significantly lower than those retrieved with climatological LER. The underestimation of OCP is mainly related to underestimation of $O_2$-$O_2$ SCDs in those areas. This issue requires further investigation. It should be noted that the cross track dependence of OCP for cloudier scenes with $f > 0.25$ is much flatter than that for lower cloudy scenes with $f < 0.25$. The cross track dependence of OCPs significantly deviates from the uniform dependence for pixels near the edges of the OMI swath. The OCPs at the edges are substantially higher than those in the near-nadir parts of the swath. This behavior of the OCP cross-track dependence is observed for both land and ocean and is not affected by the use of different surface reflectivities. This increase of OCP retrievals at the swath edges is not understood. It should be noted that similar behavior of the OCP cross track dependence is also seen in the operational OMCLDO2 v2 retrievals for low $f$.

## 3.4 Application to the OMI NO$_2$ algorithm

### 3.4.1 Spectral dependence of ECF and CRF

The OMI NO$_2$ algorithm uses the DOAS approach to fit OMI-measured spectra in the wide spectral window of 405—465 nm. Hence, the question arises as to how the value of $f$ determined at 466 nm can be representative for the entire spectral window. A simulation experiment carried out by Gupta et al. (2016) showed that the wavelength dependence of ECF is weak. In this experiment, observed TOA radiances were simulated as a sum of the clear-sky and cloudy radiances weighted with a geometrical cloud fraction. It was shown that $f$ varied only a few percent over a wide spectral range from UV to near IR. We have verified this result using OMI-observed spectra and calculating $f$ at two additional wavelengths: 405 and 435 nm. A look up table of TOA radiances at these wavelengths was generated using VLIDORT. To calculate the clear sub-pixel TOA radiance we use the climatological surface reflectivity from Kleipool et al. (2008) with linear interpolation of the spectral dependence of the surface reflectivity.

Figure 16 shows differences between the baseline $f$ at 466 nm and $f$ calculated at additional wavelengths along with the standard deviation. The differences are shown as a function of $f$ for land and ocean separately. The difference of $f$ calculated at 435 nm is obviously lower than that for $f$ calculated at 405 nm. The $f$ differences decrease with $f$ increasing due to the

gradually decreasing contribution of the clear sub-pixel to the TOA radiance, which is mostly responsible for the $f$ spectral dependence. The $f$ differences are at maximum (less than 0.02) in the middle range of $0.4 < f < 0.6$. The $f$ differences decrease with value of $f$ decreasing. However, the relative $f$ differences increase with value of $f$ decreasing because of decreasing absolute values of $f$. Over the ocean, the $f$ differences are slightly higher than those for land except for low values
of $f$. For $f < 0.1$ the $f$ differences over ocean are noticeably lower than that over land. Overall, the spectral differences of $f$ within the $NO_2$ retrieval window are small and do not exceed $\sim 0.01$ over land and $\sim 0.015$ over ocean for the most important range of $f < 0.25$.

CRF is used for calculation of trace gas AMFs in cloudy conditions. The CRF dependence on wavelength is much more pronounced than the spectral dependence of $f$ mostly due to the spectral dependence of the measured TOA radiances. For
thick cloud, CRF is simply a fraction of TOA radiance reflected by the cloud. The CRF varies with wavelength because the radiance coming from the cloud-free part of the scene is wavelength dependent. A physical interpretation of the CRF for thin clouds is not obvious. Figure 17 shows differences between CRFs calculated at 466 nm and CRFs calculated at 405 and 435 nm. Over the ocean, the $f$ differences between 466 and 405 nm are somewhat higher than those over land. For $f < 0.25$, the CRF differences do not exceed $\sim 0.07 - 0.08$ within the $NO_2$ retrieval window. Such differences are quite acceptable for purposes
of tropospheric $NO_2$ retrievals.

### 3.4.2 Cloud effects on $NO_2$ retrievals

We used the operational OMI $NO_2$ algorithm, OMNO2 version 3 (Krotkov et al., 2017), to assess how the change in surface reflectivity affects the retrievals of stratospheric and tropospheric $NO_2$ vertical column densities (VCDs). This algorithm comprises four main steps: (1) retrieval of $NO_2$ slant column densities (SCDs) by spectral fitting of laboratory-measured spectra to
the OMI-measured absorption spectrum in the range 402-465 nm (Marchenko et al., 2015); (2) Calculation of air mass factors (AMFs) using various input parameters such as viewing geometry, surface reflectivity, cloud pressure, cloud radiance fraction, and a priori $NO_2$ profile shapes; (3) removal of cross-track striping; and (4) conversion of SCDs to VCDs using AMFs and separation of stratospheric and tropospheric components (Bucsela et al., 2013). Since the retrieval of cloud parameters (cloud pressure and effective cloud fraction) is also affected by surface reflectivity, changes in surface reflectivity affect $NO_2$ retrievals
both directly, as inputs to the AMF calculation, and indirectly, through the cloud parameters.

We conducted separate $NO_2$ retrievals using climatological LER (Kleipool et al., 2008) and GLER (Vasilkov et al., 2017) and cloud parameters retrieved using the respective surface reflectivity products. Analysis of stratospheric $NO_2$ VCDs revealed that changing surface reflectivity only in $NO_2$ retrievals (i.e. no change in cloud parameters) had minor impact ($< 1\%$) in stratospheric $NO_2$ estimates. The effect was somewhat larger with the changes in cloud parameters, with the difference in
estimated stratospheric $NO_2$ VCDs reaching up to 5%. This is expected because the troposphere-stratosphere separation scheme in OMNO2 uses $NO_2$ observations from unpolluted and cloudy areas to construct the stratospheric $NO_2$ field.

The top panels in Fig. 18 and Fig. 19 show monthly mean GLER-based tropospheric $NO_2$ VCDs for July and January 2006, respectively. Tropospheric $NO_2$ exhibits strong spatial variability, with pronounced enhancements over industrial and other source regions. In addition, we observe higher tropospheric $NO_2$ in January as compared to July, a reflection of the relatively

longer $NO_2$ lifetime and shallower boundary layer in winter. In contrast to stratospheric $NO_2$ VCDs, retrievals of tropospheric $NO_2$ VCDs are very sensitive to the changes in surface reflectivity and cloud parameters. The bottom two panels of Fig. 18 and Fig. 19 show how retrievals of tropospheric $NO_2$ VCDs are affected by replacing surface reflectivity from climatological LER with GLER. Using GLER-based surface reflectivity reduces tropospheric AMFs and enhances tropospheric $NO_2$ VCDs. The impact of changes in surface reflectivity varies with the vertical distribution of $NO_2$, with the largest effects in polluted areas.

Figure 20 quantifies the percent change in tropospheric $NO_2$ VCDs as a function of $NO_2$ levels, suggesting the GLER effect on $NO_2$ retrievals can reach as much as 20-30%. Additional effects of LER changes on $NO_2$ retrievals come through changes in cloud parameters that may cause 10-15% additional changes in tropospheric $NO_2$ VCDs.

## 4    Conclusions

We have developed a new cloud algorithm based on the $O_2$-$O_2$ absorption band at 477 nm. The main features of the algorithm are (1) a new spectral fitting method of TOA radiances to derive $O_2$-$O_2$ SCDs and (2) the use of surface GLERs that replaces climatological LER data sets. This new cloud algorithm is intended for use within the standard OMI $NO_2$ algorithm and planned as a backup algorithm for the upcoming TEMPO geostationary mission.

Validation of our cloud algorithm was carried out by comparisons of the retrieved values of $f$ and OCP with values from the latest version of the OMI operational algorithm, OMCLDO2 v2, also based on the $O_2$-$O_2$ absorption band at 477 nm. $f$ and OCP were retrieved for the climatological surface LER identical to OMCLDO2. Comparisons showed a good agreement between our $f$ and that from OMCLDO2. Our OCPs are overall higher than those from OMCLDO2 by about $\sim$50 hPa on average. Diagnostic scene pressures from our algorithm are slightly closer to the surface pressure than those from OMCLDO2.

We examined $f$ and OCP changes caused by replacing the climatological surface LERs by GLERs. For the scenes with $f < 0.25$, the range traditionally used in the trace-gas retrievals, values of $f$ retrieved with GLER are higher than those retrieved with climatological LER by $\sim$0.02 on average. Even though the $f$ differences are small on average, they can be as large as 0.05-0.07 for individual pixels; this is quite significant for the low values of $f$. Over land, the GLER-retrieved OCPs are slightly higher than those retrieved with climatological LER. Over ocean, the data are closer to the 1:1 line than that over land. The geographical regions where the GLER-retrieved OCPs are lower than those retrieved with climatological LER are mostly related to sun glint areas.

We applied the new cloud algorithm to OMI $NO_2$ retrievals and analyzed $NO_2$ column changes related to the use of the new cloud algorithm. The GLER effect on $NO_2$ AMFs can increase tropospheric $NO_2$ retrievals by 20-30% over polluted regions. An effect on $NO_2$ retrievals that comes through changes in cloud fraction and pressure can make 10-15% additional changes in tropospheric $NO_2$ VCDs.

## 5 Data availability

The MODIS gap-filled BRDF Collection 5 product MCD43GF used for calculation of GLER in this paper is available at ftp://rsftp.eeos.umb.edu/data02/Gapfilled/. The OMI Level 1 data used for calculations of GLER are available at https://aura.gesdisc.eosdis.nasa.gov/data/Aura_OMI_Level1/. The OMI Level 2 Collection 3 data that include cloud, $NO_2$, and OMI pixel corner products are available at https://aura.gesdisc.eosdis.nasa.gov/data/Aura_OMI_Level2/.

*Competing interests.* The authors declare that they have no conflict of interest.

*Acknowledgements.* Funding for this work was provided in part by the NASA through the Aura science team program. We thank Pepijn Veefkind and Maarten Sneep of KNMI for providing OMCLDO2 v2 data for comparisons.

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

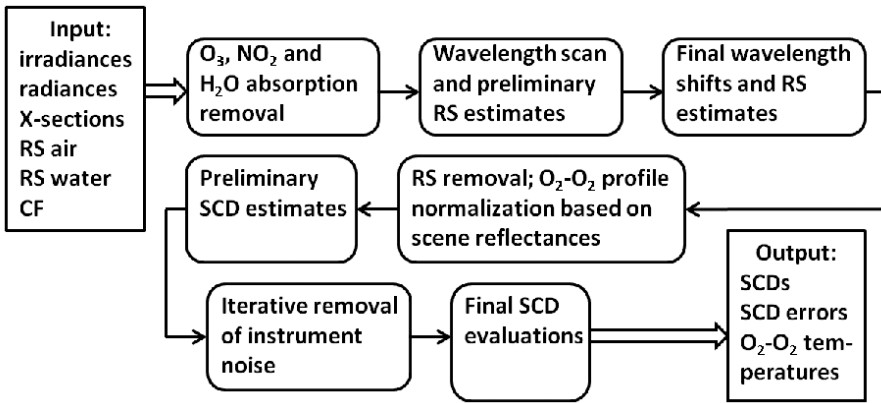

**Figure 1.** Flow diagram of the $O_2-O_2$ SCD retrieval algorithm. The algorithm input comprises: the OMI monthly-mean solar irradiances, the radiances (wavelength, line-of-sight (row) and position (along-orbit) -dependent), the laboratory cross-sections of $O_3$, $NO_2$ and $H_2O$ (X-sections), the atmospheric (RS air) and liquid-water (RS water) Raman scattering spectra (all X-sections convolved with the row- and wavelength-dependent OMI instrument line-shape functions), the OMI cloud-fraction (CF) estimates provided by an independent retrival. RS denotes the amplitudes of the combined air and water Raman scattering spectrum.

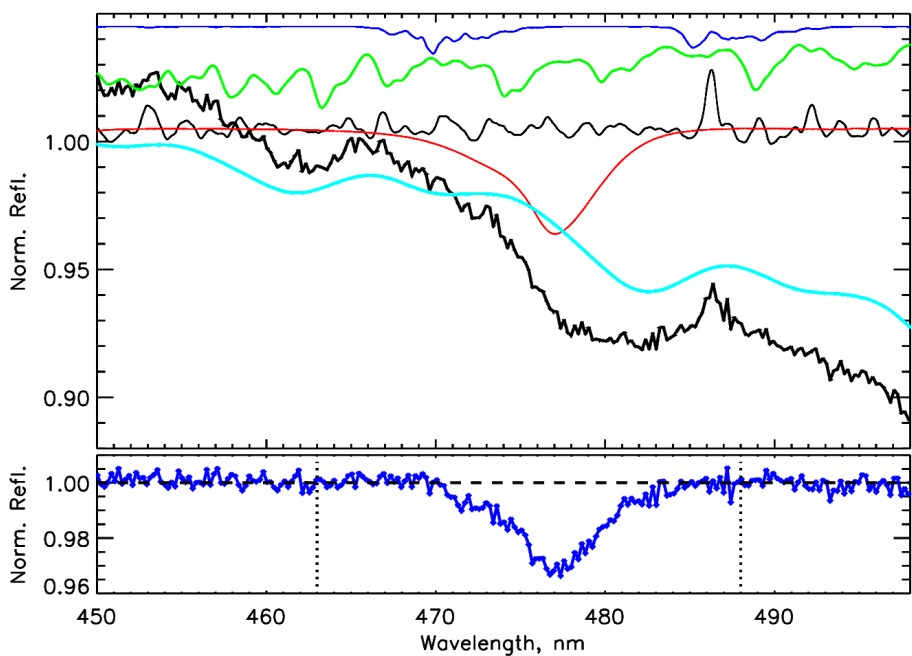

**Figure 2.** Upper panel: Reflectance normalized at $\lambda$=464 nm (bold black line) for the OMI orbit 7921 from January 10, 2006, with row 14 and orbital exposure 1550 at 65.37°N and 88.58°E (high slant column ozone values). For reference, the arbitrarily shifted and scaled absorption spectra of $H_2O$ (thin blue line), $NO_2$ (green), $O_2-O_2$ (red) and cyan ($O_3$) are plotted in the upper portion of the panel. The arbitrarily scaled and shifted Ring patterns (as seen in reflectances) are shown in black. Lower panel: the rectified $O_2-O_2$ absorption profile (i.e., the ratio of the data denoted by the red and blue lines in the upper panel, with additional adjustments to the blue-line data - see text for more details). The dashed black line shows the 1.0 reference level. In the lower panel, the vertical dotted lines denote the wavelength range used in the SCD fits of the $O_2-O_2$ absorption profile.

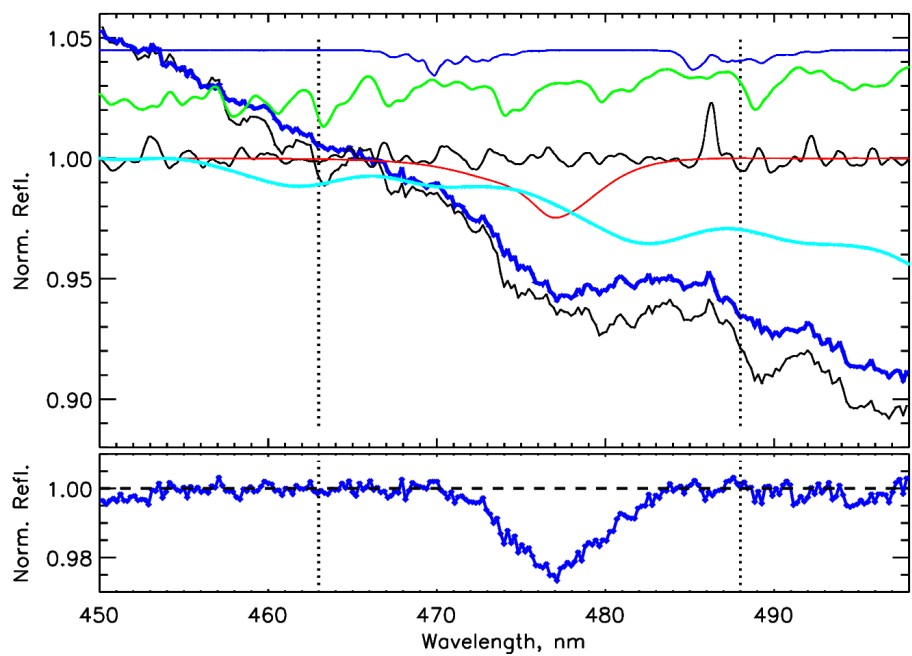

**Figure 3.** Similar to Fig. 2 but for row 44 and orbital scan number iTime=1315 pointing to the Beijing area. In the upper panel, the thick blue line follows the reflectances after removal of the trace-gas ($O_3$ and $H_2O$) absorption, however with the Ring-spectrum features remaining intact. This is to be compared to the adjascent black line that follows the original reflectances. The lower panel shows the normalised $O_2$-$O_2$ profile as used in the SCD retrieval.

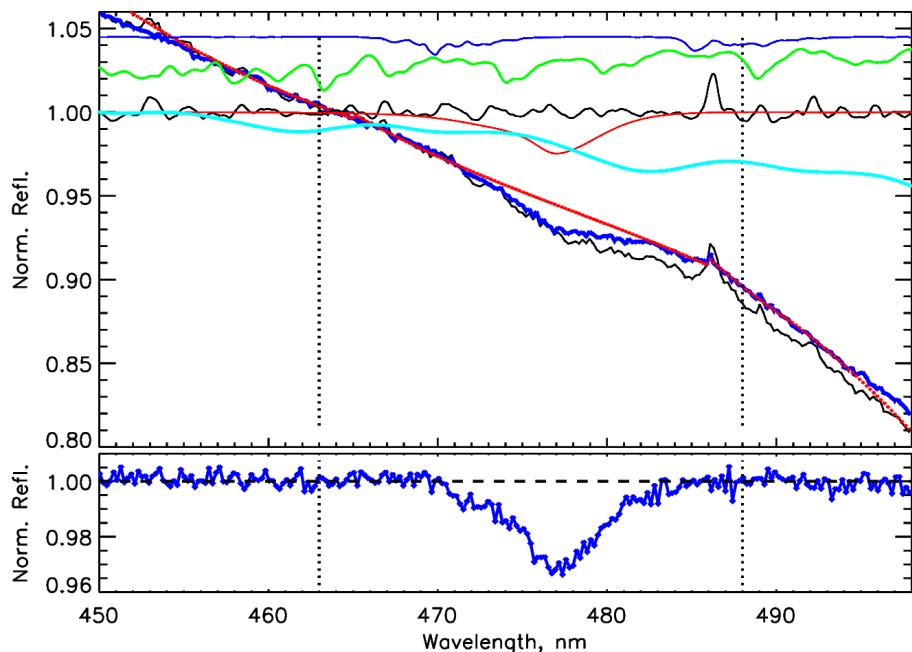

**Figure 4.** Similar to Fig. 3 but for open-water nearly cloud free ($f < 0.05$) region of the Indian Ocean (54.02°S, 106.91°E, OMI orbit 7791, January 01, 2006). The thick blue line follows these reflectances after removal of the Ring patterns and the trace-gas (O3, NO2 and H2O) absorption. The red line shows the piece-wise fit to the blue line.

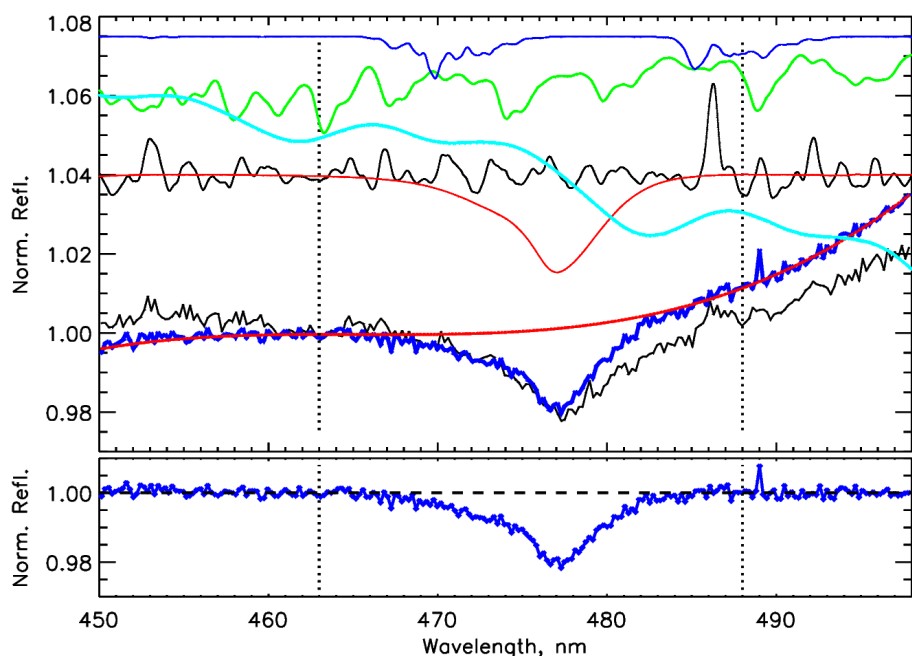

**Figure 5.** Similar to Fig. 4 but for the Sahara desert (OMI orbit 8013, January 16, 2006, row 20, orbital exposure 1180). The lower panel shows the rectified $O_2-O_2$ absorption profile, i.e., the ratio of the data denoted by the red and blue lines in the upper panel, with additional adjustments to the blue-line data - see text for more details.

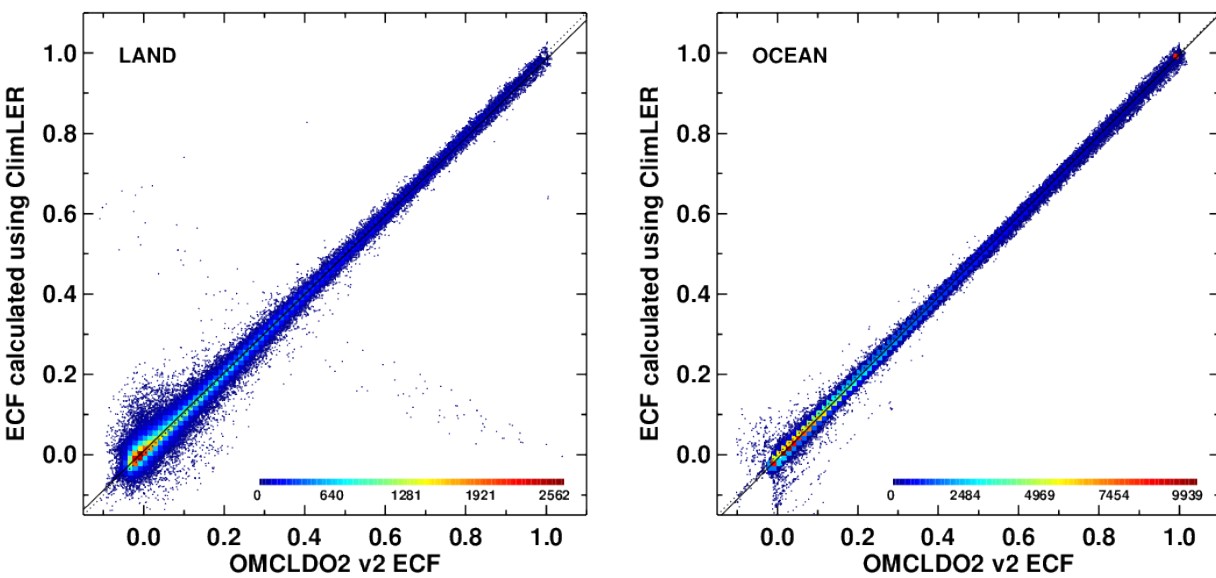

**Figure 6.** ECFs retrieved with our algorithm versus those retrieved from OMCLDO2. Results are provided as 2-D densities in ECF bins of 0.01. The color scale represents the number of OMI pixels falling withing a given bin. Data for Nov. 13, 2006, $30°$S–$30°$N. Left panel: land; Right panel: ocean.

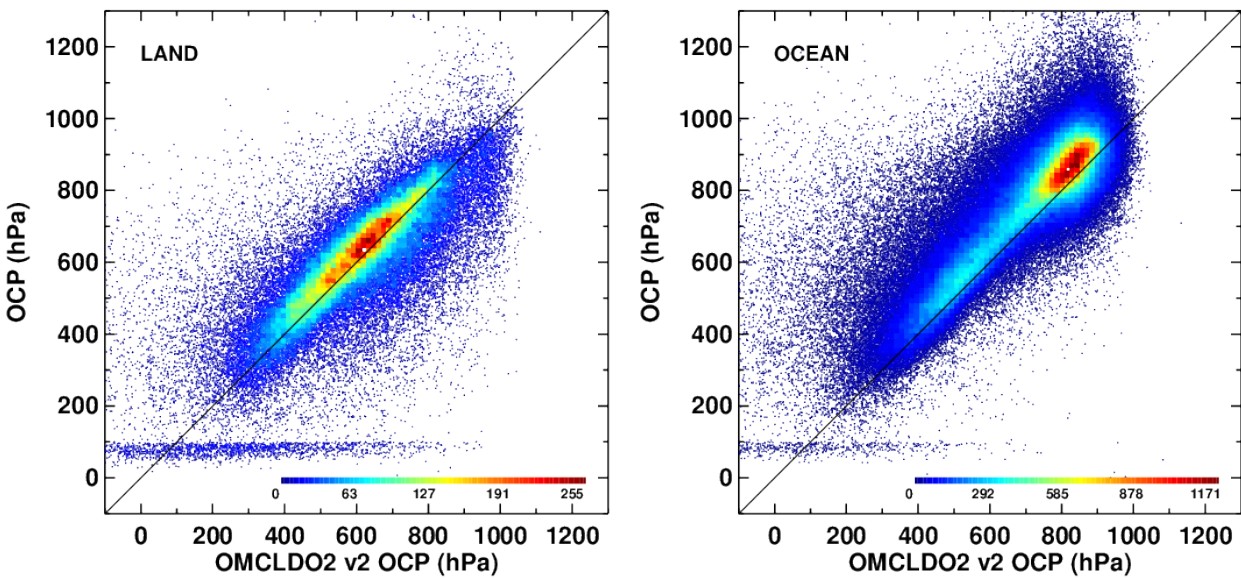

**Figure 7.** Similar to Fig. 6 but for OCP.

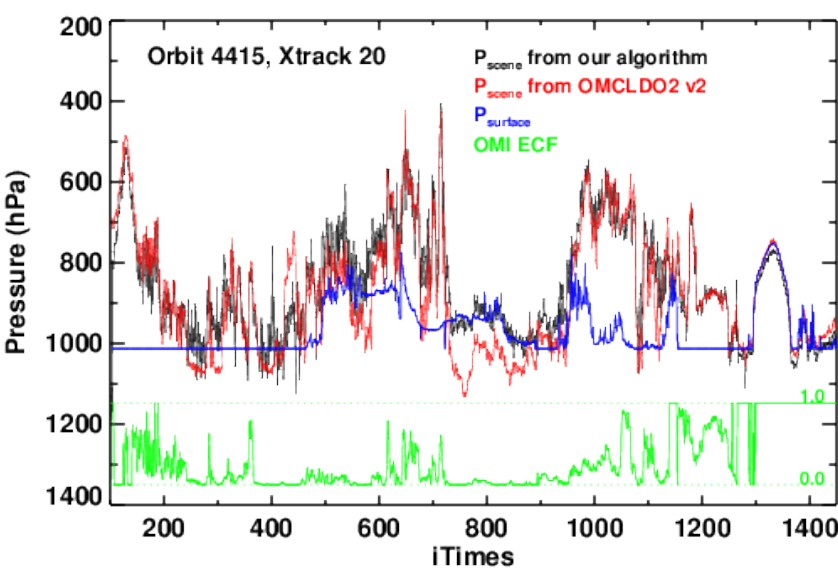

**Figure 8.** Comparison of scene pressures from our algorithm (black curve) and OMCLDO2 v2 (red) with surface pressures (blue) along cross track position 20 of OMI orbit 4415 (May 14, 2005). The green curve shows cloud fractions for this cross track position.

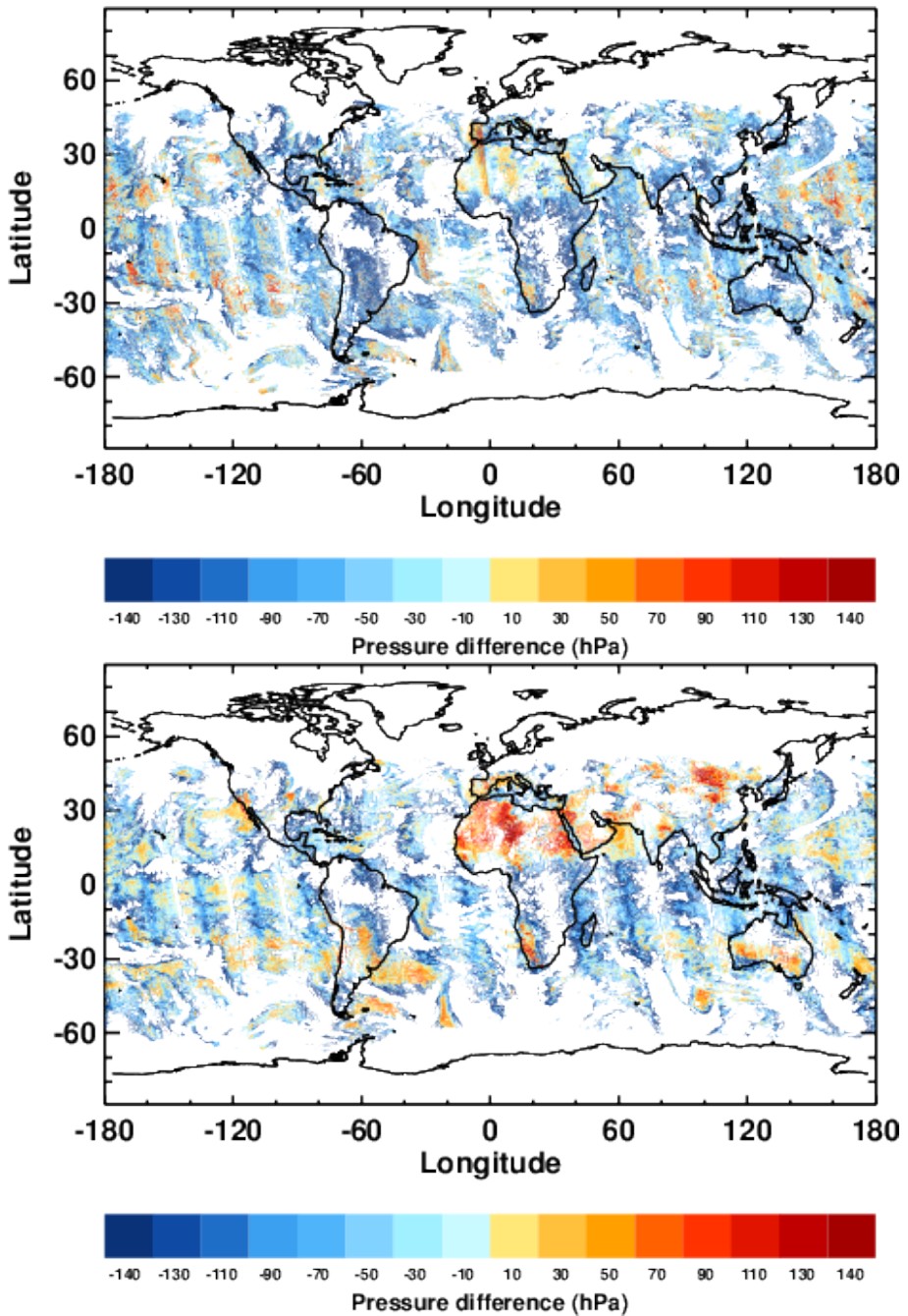

**Figure 9.** Difference between the scene pressure and the surface pressure, $P_{sc} - P_s$, for our algorithm (upper panel) and OMCLDO2 v2 (bottom panel). Data are for Nov. 13, 2006.

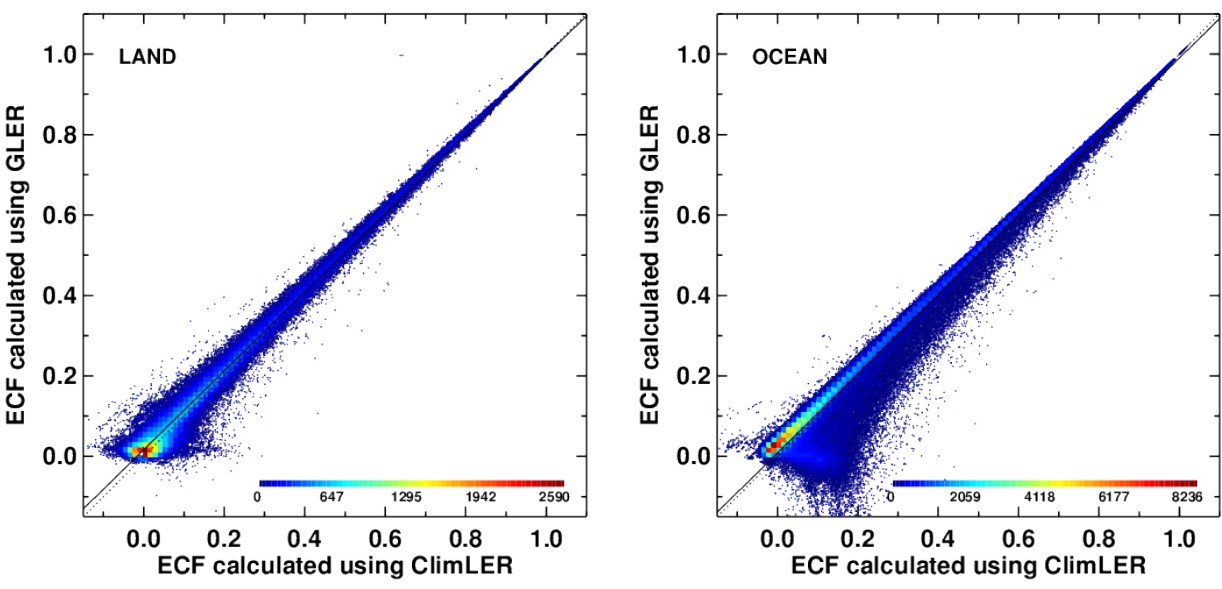

**Figure 10.** 2-D histograms comparing effective cloud fraction (ECF) retrieved with GLER (y axes) and climatological LER (x axes) for land (left panel) and ocean (right panel). The color scale shows numbers of data points. OMI data are for Nov. 13, 2006.

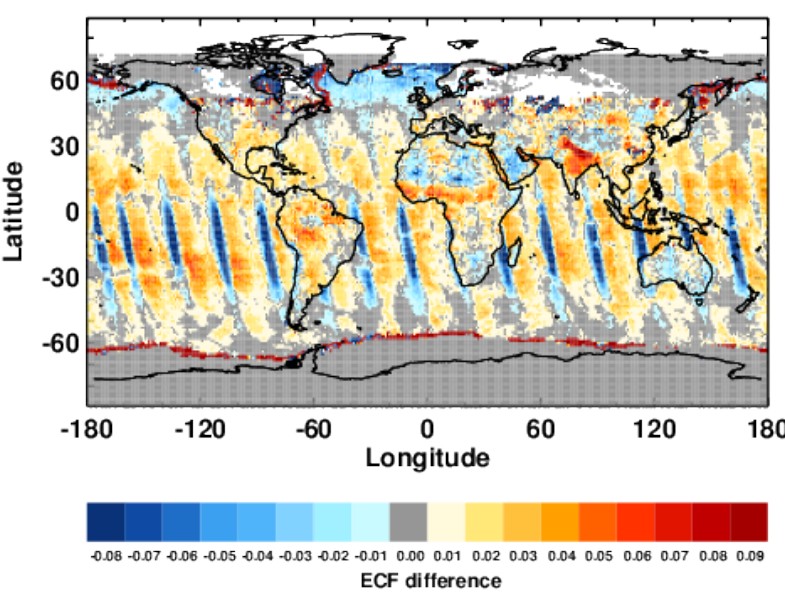

**Figure 11.** Differences between ECFs retrieved with GLER and those retrieved with climatological LER, $f(\text{GLER}) - f(\text{ClimLER})$. OMI data are for Nov. 13, 2006. No snow/ice covered areas included in the comparison.

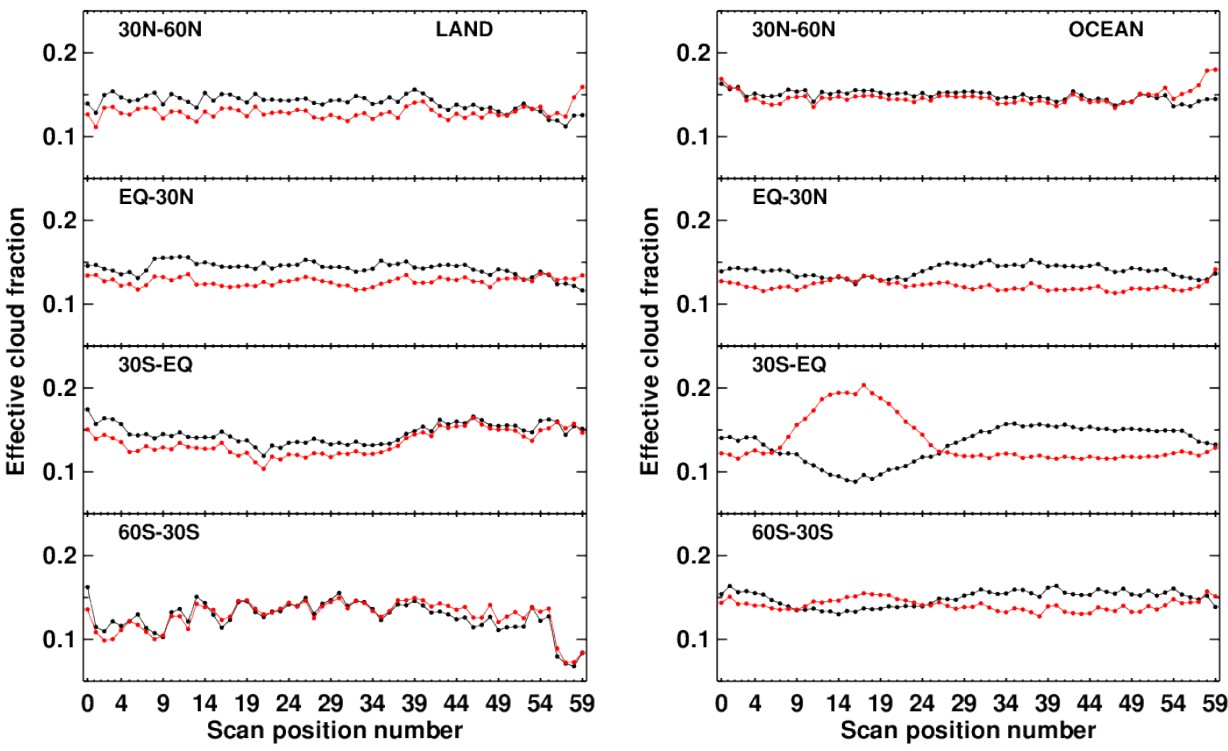

**Figure 12.** Cross-track dependence of ECF zonal means retrieved with GLER and climatological LER for different latitude bins. Data are for Nov. 13, 2006; effective cloud fractions are between 0.05 and 0.25.

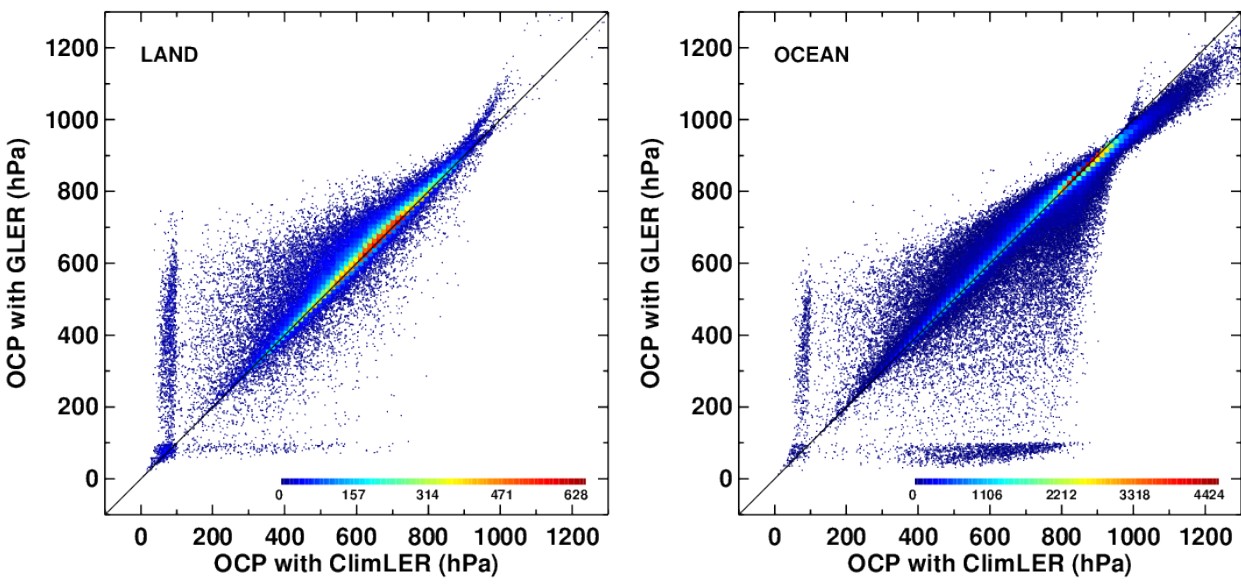

**Figure 13.** 2-D histogram similar to Fig. 10 but comparing OCPs retrieved with GLER with those retrieved with climatological LER.

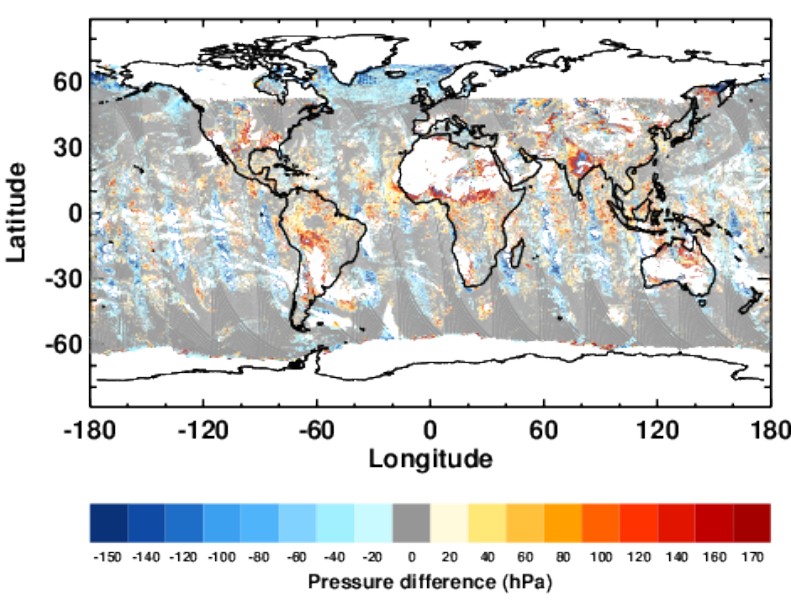

**Figure 14.** Differences between OCPs retrieved with GLER and climatological LER for Nov. 13, 2006. Data are shown for ECF > 0.05 only.

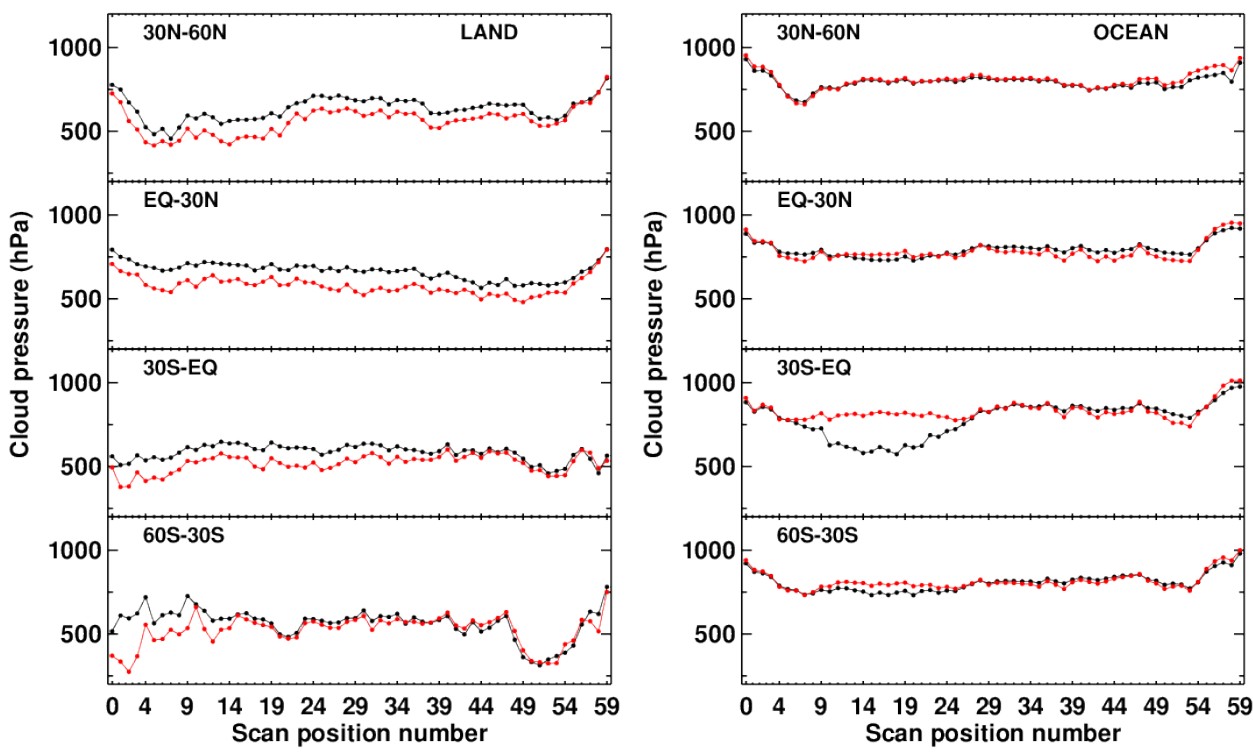

**Figure 15.** Cross track dependence of OCPs retrieved with GLER and those retrieved with climatological LER for Nov. 13, 2006.

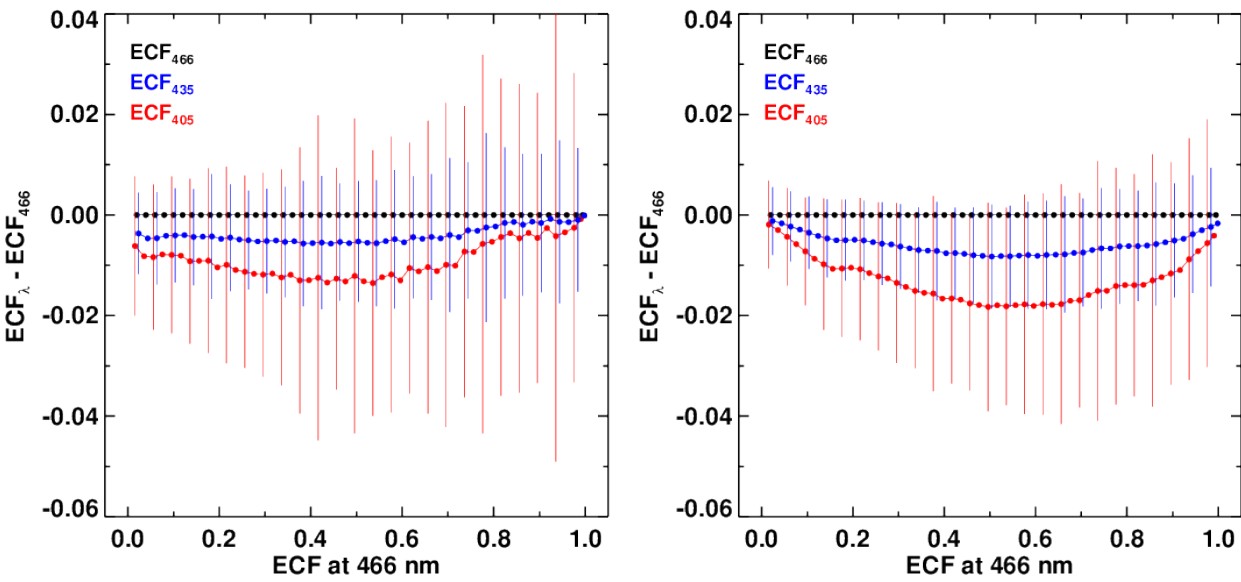

**Figure 16.** Differences between ECFs retrieved at 405 and 435 nm and ECF retrieved at 466 nm as a function of ECF at 466 nm for Nov.13, 2006. Left panel - land; right panel - ocean.

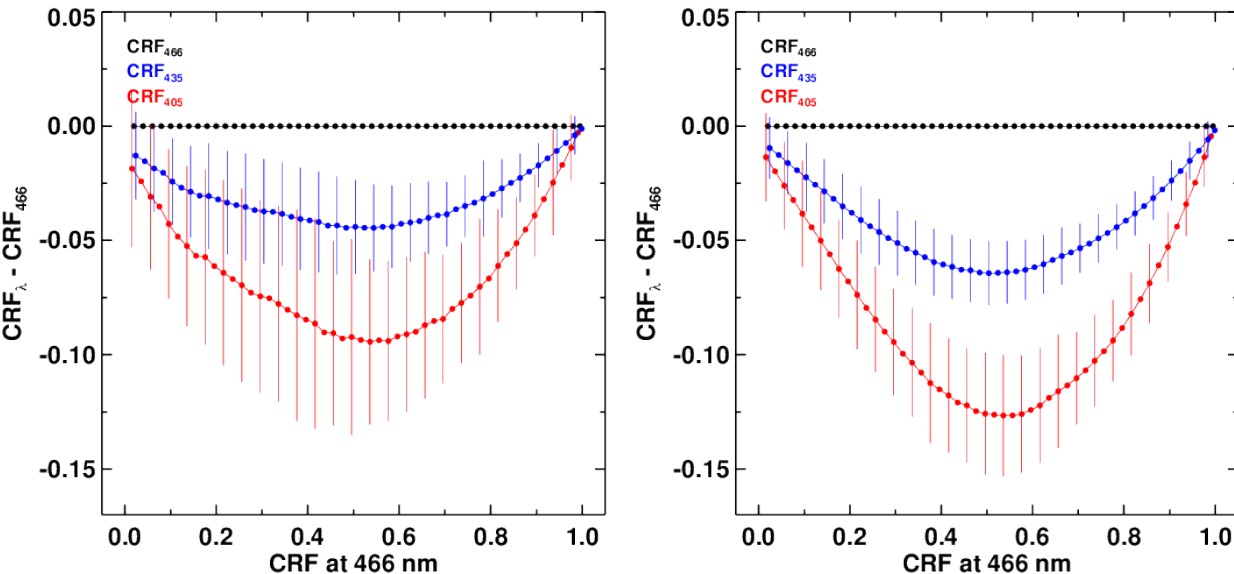

**Figure 17.** Similar to Fig. 16 but for CRF.

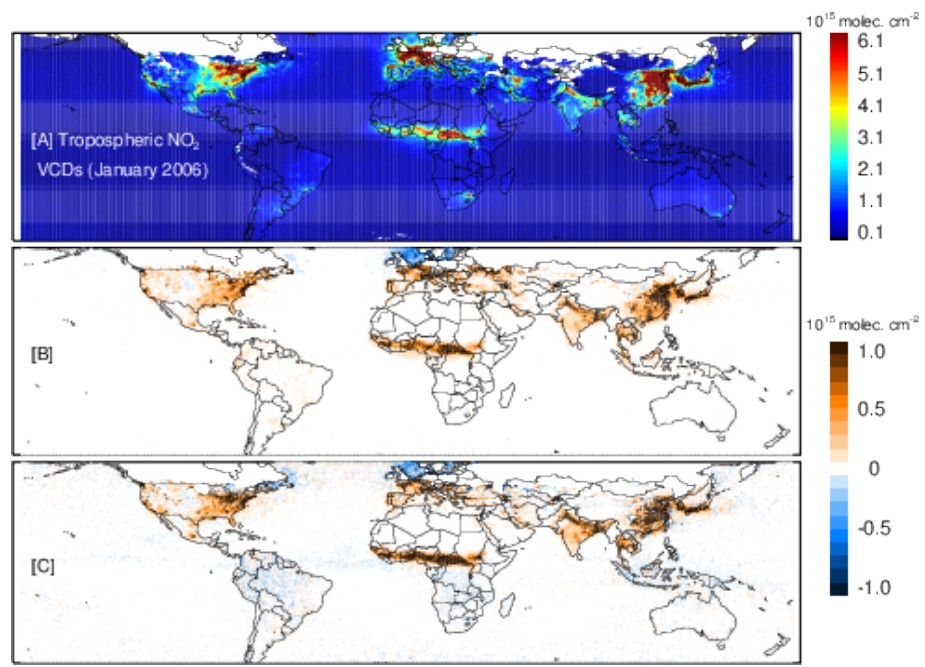

**Figure 18.** (a) Monthly-mean gridded (0.5° latitude ×0.5° longitude) OMI $NO_2$ tropospheric VCDs for January 2006 retrieved using the GLER and $O_2$-$O_2$ cloud products. (b) Change in tropospheric VCDs due to the change in surface reflectivity for $NO_2$ retrievals alone. (c) Change in tropospheric VCDs due to the change in surface reflectivity for both $NO_2$ and cloud retrievals.

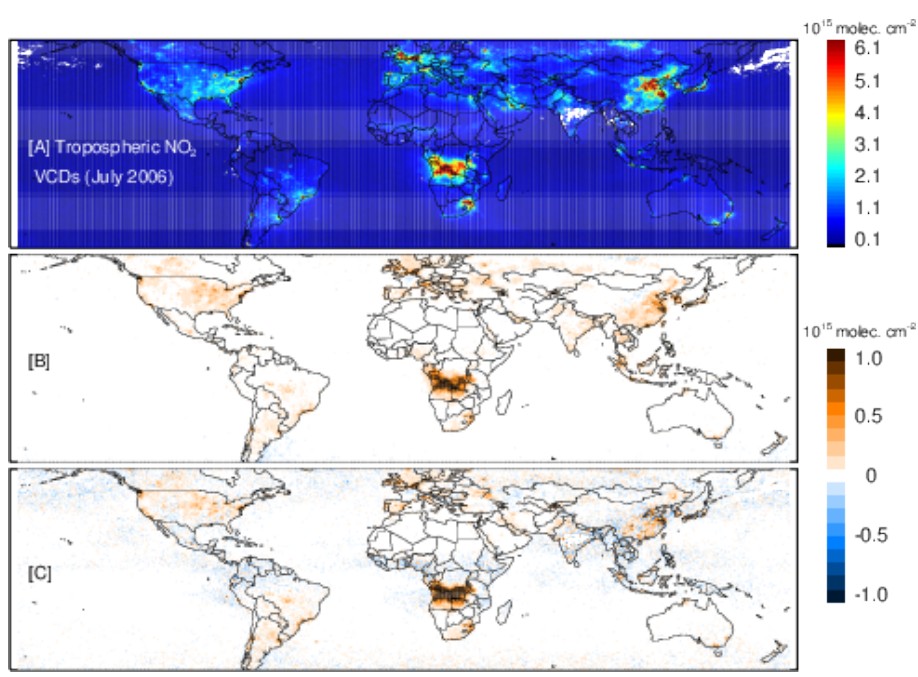

**Figure 19.** Similar to Fig. 18 but for July 2006.

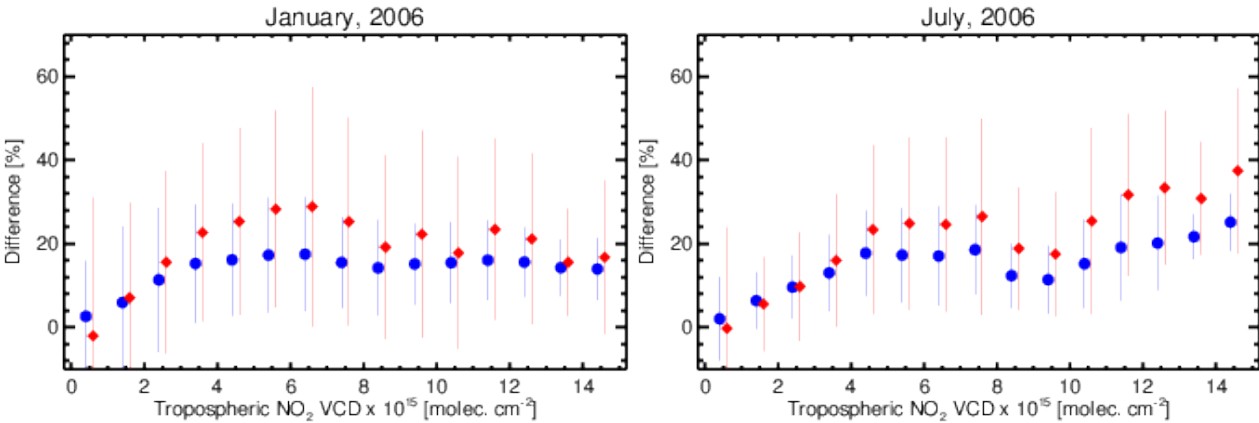

**Figure 20.** Difference in tropospheric $NO_2$ VCD caused by replacing climatological LER by GLER for July (left) and January (right), 2006. Surface reflectivity affects $NO_2$ retrievals directly as an input to the AMF calculation (blue symbols) and indirectly by changing cloud parameters used in the AMF calculation (red symbols). Vertical bars represent standard deviation for each class of tropospheric $NO_2$ VCD of size $1 \times 10^{15}$ molec cm$^{-2}$. The standard deviations contain both effects of surface reflectivity on $NO_2$ retrievals.