# Peer review of "A cloud algorithm based on the O2-O2 477 nm absorption band featuring an advanced spectral fitting method and the use of surface geometry-dependent Lambertian-equivalent reflectivity"

_Atmospheric Measurement Techniques, 2018_

## Referee Comment (RC1) · Anonymous Referee #1 · 5 May 2018

General comments:

This paper presents a cloud retrieval algorithm using the O2-O2 absorption feature centered at 477nm. I am impressed by the detailed description of the algorithm and the results. The paper is well written and relevant to the community. I recommend publication after addressing the minor issues listed below:

A general question I have is how to use this algorithm independently. As stated in the

[Figure]

paper, the retrieved cloud information is intended for the NO2 retrieval for OMI and TEMPO mission, yet the cloud algorithm needs NO2 info for the fitting process. Isn't that a paradox? OMI has the NO2, O3, and H2O retrievals from independent retrieval algorithms for the fitting process described in this paper; will TEMPO also have these trace gas retrievals as input to this cloud algorithm?

Specific comments:

Page 2, Line 5-6: " Other cloud parameters include the cloud phase, the cloud particle shape, and the particle size distribution that determine the cloud phase scattering function" is not a full sentence. I guess you want to say these parameter are not considered.

---

## Referee Comment (RC2) · Anonymous Referee #2 · 18 May 2018

This study presents an improved cloud algorithm based on the O2-O2 collision band around 477 nm, which is intended to be used for OMI and TEMPO NO2 retrieval algorithm. The improvements include a new spectral fitting approach to derive O2-O2 SCDs and the use of geometry-dependent LER (GLER) to replace climatological LER assumption. The new algorithm are clearly described in the context of retrieving NO2 from. The performance of the new algorithm is throughly evaluated by comparing retrieved effective cloud fraction (ECF) and cloud pressure (OCP) with OMI standard

algorithms. Authors also compared the retrieved ECF and OCP between the uses of GLER versus climatological LER. The results are well presented and discussed. This is overall an excellent piece of work. I only have a couple of following comments.

First, while the detailed steps of the spectral fitting and cloud algorithms (section 2.2-2.3) are well describe, I still feel difficult to follow. So, I would suggest add a flowchart of algorithm procedures, which will help readers better follow the text.

Second, (page 8, line 25) what's the reason for higher OCP retrievals based on your algorithm? I assume it is because different spectral wavelengths are used. But is that also related to the definition of cloud pressure (e.g., OMCLDO2 uses cloud top pressure?)?

---

## Author Comment (AC1) · 18 Jun 2018

Response to referee #1

We thank the reviewer for his/her evaluation of our paper and useful comments that helped improve the manuscript. We appreciate reviewer's time and effort in reviewing the manuscript. Below are our responses to each comment. Reviewer's comments are in the standard font while the responses are in the italic font.

On behalf of the authors,

Alexander Vasilkov

General comments:

A general question I have is how to use this algorithm independently. As stated in the paper, the retrieved cloud information is intended for the NO2 retrieval for OMI and TEMPO mission, yet the cloud algorithm needs NO2 info for the fitting process. Isn't that a paradox? OMI has the NO2, O3, and H2O retrievals from independent retrieval algorithms for the fitting process described in this paper; will TEMPO also have these trace gas retrievals as input to this cloud algorithm?

*This is an important question. We added the following paragraph at the end of Section 2.3.*

*As implemented, the algorithm relies on optimal SCD retrievals of the O3, NO2 and H2O trace gases, as well as preliminary cloud-fraction estimates. The latter is used exclusively over deep-water areas during the wavelength calibration and the Raman scattering (RS) removal. If needed, such cloud fractions can be substituted for appropriately adjusted reflectances, thus vying for self-sufficiency. The use of independent O3, NO2 and H2O SCDs is an essential part of the algorithm that, especially for the scenes with heavy O3 and NO2 loads, leads to more accurate O2-O2 SCDs. The use of the trace-gas SCDs does not create any paradox when the NO2 values would be used in order to retrieve cloud properties that should be incorporated into the NO2 estimates. Note that in the implemented algorithm we use the NO2 SCD estimates that can be obtained without any relevance on cloud properties. These cloud properties are used much later, during the conversion of the NO2 slant columns to the NO2 vertical columns. Opting for a complete self-reliance of the cloud algorithm, one may substitute the required O3, NO2 and H2O SCDs for SCD estimates provided by the appropriate trace-gas climatologies.*

Specific comments:

Page 2, Line 5-6: " Other cloud parameters include the cloud phase, the cloud particle shape, and the particle size distribution that determine the cloud phase scattering function" is not a full sentence. I guess you want to say these parameter are not considered.

*Thanks. We corrected the phrase:*

*Other cloud parameters: the cloud phase, the cloud particle shape, and the particle size distribution that determine the cloud phase scattering function are usually not considered.*

---

## Author Comment (AC2) · 18 Jun 2018

Response to referee #2

We thank the reviewer for his/her evaluation of our paper and useful comments that helped improve the manuscript. We appreciate reviewer's time and effort in reviewing the manuscript. Below are our responses to each comment. Reviewer's comments are in the standard font while the responses are in the italic font.

On behalf of the authors,

Alexander Vasilkov
* * *
Comments:

First, while the detailed steps of the spectral fitting and cloud algorithms (section 2.2-2.3) are well describe, I still feel difficult to follow. So, I would suggest add a flowchart of algorithm procedures, which will help readers better follow the text.

*Agree. We added the following flowchart of the SCD algorithm:*

[Figure]

*Fig. 1. Flow diagram of the $O_2$-$O_2$ SCD retrieval algorithm. The algorithm input comprises: the OMI monthly-mean solar irradiances, the radiances (wavelength, line-of-sight (row) and position (along-orbit) -dependent), the laboratory cross-sections of $O_3$, $NO_2$ and $H_2O$ (X-sections), the atmospheric (RS air) and liquid-water (RS water) Raman scattering spectra (all X-*

*sections convolved with the row- and wavelength-dependent OMI instrument line-shape functions), the OMI cloud-fraction (CF) estimates provided by an independent retrieval. RS denotes the amplitudes of the combined air and water Raman scattering spectrum.*

Second, (page 8, line 25) what's the reason for higher OCP retrievals based on your algorithm? I assume it is because different spectral wavelengths are used. But is that also related to the definition of cloud pressure (e.g., OMCLDO2 uses cloud top pressure?)?

*The definition of cloud pressure is same for both algorithms because the both algorithms are based on the MLER model (i.e. OMCLDO2 does not use the cloud top pressure). The both algorithms retrieve OCP from O2-O2 SCD estimates. However, our SCD algorithm is quite different from OMCLDO2. The reason for higher OCP retrievals from our cloud algorithm can be related to slightly higher SDC estimates and also to differences in the effective cloud fraction. We added in the text:*

*Higher OCP retrievals from our algorithm as compared to OMCLDO2 can be related to slightly higher $O_2$ - $O_2$ SDC estimates and also to differences in ECF which affect the OCP retrievals. .*